# The Impact of COVID-19 on Kidney Transplant Recipients in Pre-Vaccination and Delta Strain Era: A Systematic Review and Meta-Analysis

**DOI:** 10.3390/jcm10194533

**Published:** 2021-09-30

**Authors:** Kumar Jayant, Isabella Reccia, Piotr J. Bachul, Yaser Al-Salmay, Jordan S. Pyda, Mauro Podda, Angelica Perez-Gutierrez, Frank J. M. F. Dor, Yolanda Becker, Diego di Sabato, John LaMattina, Rolf Barth, John Fung, Piotr Witkowski

**Affiliations:** 1Transplantation Institute, Department of Surgery, University of Chicago, Chicago, IL 60637, USA; Kumar.Jayant@uchospitals.edu (K.J.); Piotr.Bachul@uchospitals.edu (P.J.B.); Yaser.Al-Salmay@uchospitals.edu (Y.A.-S.); rperezgutierrez@bsd.uchicago.edu (A.P.-G.); ybecker@surgery.bsd.uchicago.edu (Y.B.); ddisabato@surgery.bsd.uchicago.edu (D.d.S.); jlamatti@bsd.uchicago.edu (J.L.); rbarth@bsd.uchicago.edu (R.B.); jfung@surgery.bsd.uchicago.edu (J.F.); 2Department of Surgery and Cancer, Imperial College Healthcare NHS Trust, London W12 0HS, UK; isabella.reccia@nhs.net; 3Department of Surgery, Beth Israel Deaconess Medical Center, Boston, MA 02215, USA; jordanpyda@gmail.com; 4Department of Emergency Surgery, Azienda Ospedaliero-Universitaria Di Cagliari, University Hospital Policlinico Duilio Casula, 09124 Cagliari, Italy; mauropodda@ymail.com; 5Imperial College Renal and Transplant Centre, Hammersmith Hospital, Imperial College Healthcare NHS Trust, London W12 0HS, UK; frank.dor@nhs.net

**Keywords:** COVID-19, kidney transplantation, SARS-CoV-2

## Abstract

Herein, we performed a meta-analysis of published clinical outcomes of corona virus disease 2019 (COVID-19) in hospitalized kidney transplant recipients. A systematic database search was conducted between December 1, 2019 and April 20, 2020. We analyzed 48 studies comprising 3137 kidney transplant recipients with COVID-19. Fever (77%), cough (65%), dyspnea (48%), and gastrointestinal symptoms (28%) were predominant on hospital admission. The most common comorbidities were hypertension (83%), diabetes mellitus (34%), and cardiac disease (23%). The pooled prevalence of acute respiratory distress syndrome and acute kidney injury were 58% and 48%, respectively. Invasive ventilation and dialysis were required in 24% and 22% patients, respectively. In-hospital mortality rate was as high as 21%, and increased to over 50% for patients in intensive care unit (ICU) or requiring invasive ventilation. Risk of mortality in patients with acute respiratory distress syndrome (ARDS), on mechanical ventilation, and ICU admission was increased: OR = 19.59, OR = 3.80, and OR = 13.39, respectively. Mortality risk in the elderly was OR = 3.90; however, no such association was observed in terms of time since transplantation and gender. Fever, cough, dyspnea, and gastrointestinal symptoms were common on admission for COVID-19 in kidney transplant patients. Mortality was as high as 20% and increased to over 50% in patients in ICU and required invasive ventilation.

## 1. Introduction

The initial strain of coronavirus, severe acute respiratory syndrome coronavirus 2 (SARS-CoV-2), is highly transmissible and has led to a global pandemic afflicting around sixty-five million people in the first year. To date, the reported mortality rate is 2–6%, but rises to as high as 26% among hospitalized patients and represents a significant health concern in the elderly and populations with underlying comorbidities [1,2,3]. The implications of immunosuppression in the context of both SARS-CoV-2 infection and potential donor transmission are still being explored [4,5]. This uncertainty contributed to the decline in the kidney transplantation services during the initial stage of pandemic due to unquantified but potential high risk of morbidity and mortality [6,7]. On the other hand, pre-COVID-19, chronic kidney disease patients on dialysis are particularly vulnerable with an approximately ten-fold higher risk of mortality than the general population [8,9]. Consequently, Boyarsky et al. [10] recently reported a 2.2-fold (95% CI, 1.88–2.62) increase in mortality for patients in the United States on the kidney transplantation waitlist during the early part of the pandemic according to the Scientific Registry of Transplant Recipients data. Given the significant overall mortality benefit following kidney transplantation for patients with end stage renal disease (ESRD), there is unprecedented need and urgency to quantify the risk of SARS-CoV-2 infection in these patients before and after transplant [10]. The objective of this meta-analysis was to better understand the clinical course of COVID-19 in kidney transplant recipients and to identify factors associated with clinical outcomes and mortality. Furthermore, this systematic review focuses on the understanding of attributes influencing mortality following COVID-19 in kidney transplant recipients in contrast to general population and chronic kidney disease patients.

## 2. Materials and Methods

### 2.1. Search Strategy

We performed a systematic literature search of articles indexed in PubMed, EMBASE, MedRxiv, Cochrane, Crossref, Scopus, and clinical trial registries. The search strategy was based on recommendations from the Cochrane Handbook for Systematic Reviews of Intervention and reported according to the guidelines of meta-analysis of observational studies in epidemiology [11]. The following MeSH terms were used: “COVID-19” AND “kidney transplantation”; “Coronavirus” AND “kidney transplantation”; “COVID-19” AND “kidney transplantation” AND “mortality”; “COVID-19” AND “kidney transplantation” AND “clinical outcomes”. We performed additional free texts searches including renal transplantation as applicable. The initial database search was performed on 10 November 2020. Additional studies were identified via a manual search of preprints, case reports, abstracts, bibliographies, and a citation list of relevant articles using the free search terms “2019 novel coronavirus”, “SARS-CoV-2”, and “2019-nCoV infection”. The additional search was completed on 20 April 2020. This study was registered in PROSPERO, which is an international database of prospectively registered systematic reviews (CRD42020189637).

### 2.2. Inclusion and Exclusion Criteria

All observational studies available in the literature pertaining to COVID-19 and kidney transplantation were included. All other articles or publication types including editorials, letters, reviews, case reports, case series with less than five cases, and articles with duplicate data were excluded. The selected studies were reviewed and the following parameters were extracted: clinical presentation, severity of respiratory disease, hospital admission, rate of intensive care unit (ICU) admission, diagnosis of acute respiratory distress syndrome (ARDS), need for mechanical ventilation, presence of acute kidney injury (AKI), need for renal replacement therapy (RRT), changes in white cell count (WBC), laboratory evidence of inflammation (e.g., ESR, CRP), modifications to the immunosuppressive regimen, whether other treatment was administered, presence of graft rejection, and mortality. Furthermore, since older age, male gender, and impact of time since transplantation were evaluated to determine the mortality risk in regard to these factors [12,13]. In addition, mortality risk was computed following COVID-19 in a setting of chronic kidney disease (CKD) patients on waitlist/dialysis or both with respect to kidney transplant recipients. Preferred Reporting Items for Systematic Reviews and Meta-analyses (PRISMA) guidelines were used to complete the search and the article selection. Figure 1 demonstrates the PRISMA flowchart and identifies the number of search results, articles meeting criteria, and articles selected for data extraction.

### 2.3. Data Extraction

Two separate physician reviewers, KJ and IR, employed a two-stage method to independently screen the identified articles using a shared online form. During the initial review, the titles and abstracts were evaluated for the purpose of excluding ineligible articles. During the secondary review, the full or limited text (e.g., posters) were reviewed. In case of a discrepancy between the two initial reviewers, the items were discussed until a consensus was achieved with a third reviewer serving as the arbitrator (PW). Pre-defined data were extracted from the articles meeting criteria into a dataset and preliminary descriptive statistical analysis was performed to generate central tendency (i.e., mean or median) and dispersion (i.e., 95% CI, IQR or range) for continuous variables. In scenarios where reported means and standard deviations of the variables of interest were not available (i.e., not reported in the identified articles), the values were inputted from the reported statistics (i.e., median, IQR or range) [14,15]. Heterogeneity among included studies was investigated using I2-statistics and classified in the following fashion: I2 of ≤25% was interpreted as low heterogeneity; I2 of 25–75% indicated moderate heterogeneity, and ≥75% was interpreted as higher heterogeneity [16]. Due to the heterogeneity of data within individual studies, and between different studies, the pooled prevalence of all attributes was calculated using the random-effect model in STATA/SE 16 (Stata, College Station, TX, USA). The risk of bias for observational studies was appraised through the quality assessment tool published by the National Institutes of Health (i.e., Quality Assessment Tool for Case Series Studies) [17], which is the preferred tool for assessments of risk of bias in systematic reviews registered in the PROSPERO protocols (Figure 2) [18].

## 3. Results

### 3.1. Search Results

The primary literature search yielded a total of 216 articles meeting the preliminary selection criteria. After careful evaluation, 168 articles were removed based on the exclusion criteria as detailed above. In the case of duplicate publications, the most recent available data were considered and included in the analysis. After the resolution of differences between reviewers and removal of duplicate reports, all available observational retrospective studies and case series were selected for data extraction [19,20,21,22,23,24,25,26,27,28,29,30,31,32,33,34,35,36,37,38,39,40,41,42,43,44,45,46,47,48,49,50,51,52,53,54,55,56,57,58,59,60,61,62,63,64,65,66]. This process identified 48 individual studies reporting a total of 3137 renal transplant patients with COVID-19 from 12 different countries located in North America, Europe, and Asia (Table A1, Table A2 and Table A3). The detailed results of the meta-analysis are presented below. 

### 3.2. Demographic Data

The pooled estimated mean age of the patients was 57.08 years (95% CI 54.55–59.03 years). The gender distribution in the reported data population was 66% (95% CI 61–70%) male. The post-transplant period was reported in 33 studies with a mean of 7.06 years (95% CI, 5.91–9.02 years). Of these, 17 studies provided sufficient data to assess the time between kidney transplantation and the diagnosis of COVID. Overall, 66.3% (827/1248) of COVID-19 positive patients were more than 1–2 years post kidney transplant and 33.7% (421/696) were within 1–2 years of transplantation. Duration of COVID-19 symptoms before hospitalization was inconsistently reported and thus not analyzed.

### 3.3. Comorbidities

Meta-analysis utilizing a random-effects model was performed in order to identify the effect size (ES) of comorbidities in kidney transplant recipients with confirmed COVID-19 and expressed as a percentage. The estimated pooled prevalence of individual comorbidities is presented in Table A4. Type 2 diabetes mellitus (T2DM) was reported in 33 studies, in 621 out of 1785 patients with COVID-19 equating to a pooled prevalence of 34% (95% CI, 29–40%). Hypertension was determined from data available in 32 studies, with 1404 cases of hypertension out of 1733 COVID-19 patients, resulting in a prevalence of 83% (95% CI, 78–88%). Overall, the prevalence of COVID-19 was 23% (95% CI, 17–28%) in kidney transplant recipients with underlying heart disease. Based on 14 studies, the estimated pooled prevalence of obesity was 36% (95% CI, 26–46%); positive smoking status 15% (95% CI, 10–20%); chronic lung disease 15%, (95% CI, 5–24%); and malignancy 8% (95% CI, 6–11%).

### 3.4. Clinical Characteristics and Laboratory Results

The pooled estimate, with 95% CI was determined via random-effect analysis of the reported proportions. The most predominant clinical feature reported during hospitalization was fever, which was identified in 77% (95% CI, 72–81%) of patients, followed by cough 65% (95% CI, 61–69%), dyspnea 48% (95% CI, 42–53%), and gastrointestinal symptoms 29% (95% CI, 25–34%). Several articles have noted that higher levels of inflammatory markers equate with increased disease severity and a poor prognosis in COVID-19 infected patients [67]. The pooled proportion of patients with lymphocytopenia (<1000 cells/mm^3^) and high CRP (>5 mg/dL) was 79% (95% CI, 70–89%) and 60% (95% CI, 35–84%), respectively. The pooled prevalence of radiological evidence of pneumonia was documented in 81% (95% CI, 74–88%) of patients (Table A4).

### 3.5. Immunosuppression Modulation and COVID-19 Drug Treatment

The pooled prevalence of patients who received hydroxychloroquine and tocilizumab was 73% (95% CI, 66–80%) and 21% (95% CI, 16–26%), respectively (Table A4). Data from relevant studies reporting modifications to immunosuppression were extracted and analyzed. Mycophenolate mofetil/mycophenolic acid (MMF/MPA) was the most commonly withheld or reduced immunosuppressive medication in 91% (95% CI, 88–94%) of patients followed by CNI 43% (95% CI 30–56%). Additionally, pulse steroids were administered in 40% (95% CI, 30–50%) of patients.

### 3.6. Disease Severity and Mortality

Based upon data available in the analyzed studies, the pooled prevalence of COVID-19 associated hospital mortality was 21% (95% CI, 19–24%) for inpatients (718 of 3137) with a significantly higher mortality of 53% (95% CI, 44–63%) among patients admitted to the intensive care unit (ICU) (184 of 355) (*p* < 0.0001) (Figure 3a,b). An ICU admission was required in 26% (95% CI, 22–30%) of kidney transplant recipients hospitalized with COVID-19, albeit data might not be representative of the actual requirement of ICU-level care due to limited availability in the midst of the global pandemic (Figure 3c). The pooled prevalence of ARDS and AKI in COVID-19 infection was 58% (95% CI, 48–69%) and 48% (95% CI, 43–53%), respectively (Figure 3d). Similarly, the pooled prevalence of mechanical ventilation was 24% (95% CI, 20–28%) (Figure 3e) and of dialysis was 22% (95% CI, 16–29%). Furthermore, the mortality among those who required mechanical ventilation was 68% (95% CI, 58–79%) for inpatients (206/285) (Figure 3f).

Moreover, a group-wise comparison was made to determine the association of mortality and ARDS in the index population. The analysis revealed significantly higher risk of mortality in COVID-19 patients with ARDS in contrast to patients without ARDS (OR = 19.59; 95% CI, 6.64–57.78) (Figure 4a). Similarly, our analysis showed a significantly higher mortality risk in kidney transplant recipients on mechanical ventilation following COVID-19 (OR = 3.80; 95% CI, 2.35–6.14) (Figure 4b). Furthermore, our analysis demonstrated a significantly higher mortality risk in patients’ receiving ICU care (OR = 13.39; 95% CI, 7.27–24.68) (Figure 4c).

Additionally, a group-wise comparison of mortality was conducted for the following attributes: age (≥60 years vs. <60 years), time since kidney transplantation (<2 years vs. ≥2 years), and reported gender (male vs. female). The odds ratio of COVID-19 related death was significantly higher in the age group ≥60 years (OR = 3.90; 95% CI, 2.56–5.94); however, neither such association existed for time since transplant (i.e., ≥2 years) (OR = 1.37; 95% CI, 0.72–2.58) nor for gender (OR = 0.71; 95% CI, 0.50–1.01) (Figure 5a–c).

Based upon the availability of data in the included studies, comparative mortality risk following COVID-19 was assessed in CKD patients on the waitlist or on dialysis in comparison to kidney transplant recipients and analysis demonstrated no significant increased risk OR = 1.24 (95% CI, 0.92–1.66) (Figure 6).

## 4. Discussion

Kidney transplantation provides benefits of improved quality of life and reduced mortality compared to other renal replacement therapies [68,69,70]. Since the beginning of the pandemic, there has been sense of disquietude that kidney transplant recipients might predispose to abysmal outcomes after SARS-CoV-2 infection and as the pandemic evolved, most policy-makers have advised minimizing social contacts. A detailed understanding of COVID-19 susceptibility, disease process and outcomes has been essential to create a customized health advice to this subset of population.

First, transplant-associated immunosuppression renders patients generally at an increased risk of infection. Early reports suggest that risk is also true for SARS-CoV-2. Second, SARS-CoV-2 has been shown to have an affinity for angiotensin converting enzyme 2 (ACE2) receptors, which are predominantly found in the lungs and kidneys and this may be partially responsible for organ injury secondary to the infection. Additionally, uncertainty and fear of the sequalae of COVID-19 infection have resulted in a significant decline in renal transplantation, particularly during the initial stage of the pandemic [4,5].

The exact mechanism by which the viral infection causes kidney injury has not been elucidated. One leading proposed mechanism is that the virus causes direct damage to the renal tubular epithelium and podocytes via an angiotensin converting enzyme 2 mediated process, which leads to a cascade of mitochondrial dysfunction, acute tubular necrosis, formation of protein resorption vacuoles, protein leakage in Bowman’s capsule, and collapsing glomerulopathy [71,72]. Viremia may also directly lead to the damage of endothelial cells in the kidney [73]. The severe inflammatory response in the setting of dysregulated immune activation and a cytokine syndrome following SARS-CoV-2 infection might prompt acute respiratory syndrome, septic shock, and multi-organ failure including acute renal failure [74]. Other contributing factors include macrophage activation syndrome, endothelilitis, and hypercoagulability, leading to microthrombus formation, micro-embolism, and rhabdomyolysis [75].

Some insights into the mechanism of injury may be elucidated by analysis of the clinical evolution of infected patients. According to the WHO–China joint commission report on COVID-19 in the general population, fever (88%) and cough (68%) are reported to be the most common initial symptoms [76]. Interestingly, despite a decreased immune response secondary to chronic immunosuppression, we observed a similar rate cough (65%) among kidney transplant recipients with COVID-19. However, our analysis revealed a lower prevalence of fever (77% vs. 88%), but higher prevalence of dyspnea (48% vs. 18.6%) and gastrointestinal symptoms (29% vs. 3.7%) when compared to the general population studied in the WHO report [76]. A plausible explanation of the lower prevalence of fever could be because of the immunosuppressive state, whilst higher prevalence of gastrointestinal symptoms in the index cohort might be due to an exacerbation of the known toxic effects of antimetabolites on the gastrointestinal tract. Furthermore, the increased frequency of dyspnea in kidney transplant (KT) recipients could be compounded by the 10-fold higher prevalence of chronic lung disease (15%) among KT recipients compared to 1.5% in the general population with COVID-19 [77].

The mortality rate of SARS-CoV-2 in hospitalized patients as reported in recent meta-analysis of 14 studies was 4.8% [13]. Importantly, several risk factors for increased mortality have been identified including male sex (ORs = 1.50), advanced age (ORs = 4.59), hypertension (ORs = 2.70), cardiac disease (CVDs) (ORs = 3.72), T2DM (ORs = 2.41), lung disorders (ORs = 3.53), or malignancy (ORs = 3.04) [13]. The observed mortality rate among kidney transplant recipients hospitalized due to COVID-19 in our meta-analysis was 21%; although there have been wide inconsistencies throughout the literature regarding mortality data as several studies have been published upon in-complete follow-up, hence mortality figures could be higher and future studies with complete follow-up are required to completely elucidate the disease process.

Multitudes of studies have outlined the association of higher case-fatality with COVID-19 to increased prevalence of comorbidities [78,79,80]. Yang et al. performed a meta-analysis of seven studies including 1576 patients with COVID-19 and determined that the most common comorbidities were hypertension (21.1%), diabetes (9.7%), cardiovascular disease (8.4%), and chronic lung disease (1.5%) [77]. Our meta-analysis demonstrated over ~3–4 fold higher prevalence of these comorbidities among kidney transplant recipients and COVID-19 with numbers being 83% for hypertension, 34% for T2DM, 23% for cardiac disease, and 15% or ~10 fold higher for chronic lung diseases. When considering the attributed risk of age, our analysis revealed a statistically significant heightened risk in the elderly population (OR 3.90) with no increased risk when comparing sex and time since transplant after which infection was acquired. Thus, this group remains a vulnerable population due to increasing age and associated comorbidities.

Some reports have demonstrated that the natural course of COVID-19 may lead to multiorgan failure in certain subsets of the population. with regard to renal injury, a recent meta-analysis reported that 17% (0.5–80.3%) of patients had acute kidney insufficiency (AKI) with 5% (0.8–14.7%) requiring hemodialysis (HD) [81]. However, the observed prevalence of AKI and HD requirement in our study was much higher (i.e., 48% and 22%, respectively). These findings are consistent with factors specific for kidney transplant recipients including lower functional reserve of kidney allograft, and the toxic effect of tacrolimus in combination with increased susceptibility to prerenal causes of renal dysfunction (dehydration, hypotension, and metabolic disarray), which are absent in the general population.

For COVID-19 patients, clinical outcomes in kidney transplant recipients are worse than in non-transplant patients overall, however, they seem to be comparable to patients with end stage renal disease (ESRD). Two studies from the same medical center in New York City, NY reported a comparable case-fatality rate of 25% for both ESRD or kidney transplant recipient COVID-19 positive patients [82,83].

The high comorbidity rate in ESRD patients and kidney transplant recipients are well-recognized attributes, resulting in a similarly elevated risk of acute organ injury during SARS-CoV-2 infection, and also a consequently higher mortality risk than the general population [78,84,85,86]. A retrospective study involving 3988 critically ill patients from Lombardy, Italy reported increased mortality in CKD patients (i.e., 41 (78.8%) out of 52 patients and univariate analysis revealed HR 2.78 (95% CI; 2.19–3.53) [87]), although our meta-analysis demonstrated no significant increased mortality risk (OR = 1.24; 95% CI, 0.92–1.66) when both were compared. Fortunately, this observation demands further studies as recent study from NHS England “OpenSAFELY” has outlined causal factors for 10,926 COVID-19 related deaths and reported CKD as an important attribute for mortality. Furthermore, when the data from the CKD subgroups were compared, ESRD and CKD at stage 5 (HR 2.52) were found to be associated with higher COVID-19 related mortality than other isolated chronic ailments such as hypertension, obesity, cardiac disease, or chronic respiratory disease [88]. However, the risk of contracting SARS-CoV-2 in patients attending dialysis sessions three times a week (and cumulative risk from inherent comorbidities of not being transplanted) needs to be weighed against risk of immunosuppression, especially in acute transplantation. For the latter, the risk of surgery, cardiovascular mortality, infection, and increased immunosuppression burden will be elevated for some months before reaching equipoise.

A summary of studies outlined outcomes in critically ill COVID-19 patients and reported a higher mortality for the non-transplanted patients. The mortality rate among over 10,000 critically ill non-transplant patients was reported to be as high as 41% in a meta-analysis of 24 studies by Armstrong et al. [89]. However, critical care unit mortality for patients with similar characteristics in a transplant cohort reached as high as 53% with a stepwise increase in frequency of ICU admission (26%), invasive ventilation (24%), and mortality of 68% whilst on a mechanical ventilator. The estimated prevalence of ARDS in our meta-analysis was 58% (95% CI, 48–69%) and 24% of KT patients required invasive ventilation, which was comparable to the non-transplant patient cohorts in which 41.8% of 201 hospitalized COVID-19 patients developed ARDS [85] with the number rising to 68% in critical care settings [79]. Thus, the trajectory of mortality in kidney transplant recipients showed a significant upward trend, with those requiring critical care support and was further high in patients with respiratory failure, although explicit details underpinning this observation remain elusive. However, it is plausible that the hypercoagulable state of CKD along with pulmonary thromboembolic disease of critical COVID-19 are mounting the risk of thrombotic complication and increased mortality. Hence, enhanced thromboembolic prophylaxis should be considered in hospitalized cohorts of COVID-19 patients, although at present, wide variations in clinical guidelines exist and unified disease management protocol with risk stratification are yet to be established.

Inefficiency of the health system in response to the initial face of the pandemic, in addition to major concerns about the detrimental effect of the immunosuppression on a clinical course of COVID-19, has led to halting kidney transplantation procedures in most programs globally. However, delaying kidney transplantation also has negative consequences. A recent study reported a 2.2-fold rise in mortality rate in CKD patients on the wait-list for transplantation during the ongoing pandemic [10]. A study of the Spanish Society of Nephrology Registry reported higher rates of SARS-CoV-2 infection and a 23% mortality rate among ESRD patients on hemodialysis [90]. Moreover, no direct evidence is available to suggest that immunosuppression has a negative effect on the clinical course of COVID-19 in kidney transplant recipients and hence, despite the on-going pandemic, clinical transplantation services around the world have largely resumed following the initial interruption. Of note, most of the reports included in our metanalysis presented outcomes during the first phase of the pandemic and the high mortality rates reported may also be related to the transient but drastic limitations of vital resources like critical care capacity [89]. At the same time, COVID-19 vaccines were not available and the Delta variant not described. Therefore, our study provides insight on the impact of COVID-19 on kidney transplant population in such conditions. Nevertheless, it provides valuable information and may serve as a reference for future studies assessing the impact of patient vaccination as well as the Delta variant of COVID-19 in the kidney transplant population in the future.

At the beginning of the pandemic, SARS-CoV-2 infection was thought to pose a significant challenge for kidney transplant recipients due to the immunosuppressive state; however, the current understanding suggests that the immunosuppression by itself does not seem to confer an increased risk of poor outcome in SARS-CoV-2 infection, which is consistent with other observations. First, typical immunosuppression for kidney transplant recipients primarily limits the adaptive immune response rather than the innate response, with the latter seeming primary in determining COVID-19 outcome. The only form of immunosuppression strongly linked with poor outcomes identified to date is of specific deficits in the type I interferon innate response. Second, reactive corticosteroid immunosuppression with high-dose dexamethasone reduces mortality in severe COVID-19. Third, as noted by others, immunosuppression, did not emerge as a risk factor for poor outcome in SARS-CoV-1 or Middle Eastern Respiratory Syndrome infections, whereas comorbidities were analogous to those conferring risk in SARS-CoV-2 infection [91,92,93]. Interestingly, the fact that kidney transplant recipients have similar outcomes to the matched general population is in stark contrast to the very high rates of mortality reported in those with ESRD. This finding might suggest that, in the context of COVID-19, risk of ESRD and associated mortality has a more deleterious effect than pharmacological immunosuppression. It therefore remains the case that major international guidelines currently recommend against the routine cessation or reduction in immunosuppressive therapy in KT recipients before any SARS-CoV-2 infection and that modification following SARS-CoV-2 be considered under special circumstances such as in superadded bacterial infection or worsening respiratory failure. Consequently, subsequent CD8+ T cell and B cell recovery has been observed with progressive resolution of the disease [94,95]. Hence, some degree of reinstatement of a functioning immune system may be desired in KT recipients in order to limit the severity of host response to SARS-CoV-2 infection. Neither the lymphopenia observed in many patients with COVID-19 nor the established practice of antimetabolite dose reduction or even cessation during the treatment of other viral infections in transplant patients (cytomegalovirus, BK virus infections) have led to increased rates of rejection [96,97,98,99]. Recent reports have recognized the utility of steroids in critically ill COVID-19 patients, which may limit the severity of ARDS and the associated cytokine storm [100,101,102]. Consequently, the RECOVERY trial outlined the benefit of high dose dexamethasone in critically ill COVID-19 patients on mechanical ventilation [103]. Thus, there is some basis for the administration of pulsed steroids doses as was given to 41% of the patients in our meta-analysis. There are few limitations of our meta-analysis. The relatively large number of small population studies has made our composite data heterogeneous despite using a random effect model for analysis. Additionally, differences in reported clinical trajectories, management approach, and outcomes is likely to be significantly dependent on the available resources and other variable factors such as admission criteria across the different health care facilities reporting the cases.

The strengths of this meta-analysis are the extensive and exhaustive nature of the search, the independent process of study selection and data abstraction, and the random effects model used for the analysis of observational studies. We hope that this meta-analysis will contribute to a more detailed understanding of the SARS-CoV-2 disease process in the KT patient and guide future clinical decision making for this patient population.

In summary, the clinical presentation of COVID-19 in kidney transplant recipients seems to differ from the general population with lower prevalence of fever and a higher prevalence of dyspnea and gastrointestinal symptoms. The clinical course of COVID-19 in kidney transplant recipients revealed higher mortality in different settings of health care facilities including hospital, ICU, and those requiring invasive ventilation. within the cohorts’ studies of kidney transplant recipients, age and presence of multitudes of comorbidities were strongly associated with complicated disease course (i.e., increased prevalence of ARDS, AKI, and HD requirement and mortality). Whilst data regarding impact of SARS-CoV-2 in this cohort are still accumulating; however, to date, it seems clear that weight of comorbidity are the most important attributes in determining the outcome than transplant surgery alone. As being in an immunosuppressed state constitutes less risk from SARS-CoV-2 than ESRD patients on hemodialysis, by that mean, we advocate for the continuation of kidney transplantation programs.

## Figures and Tables

**Figure 1 jcm-10-04533-f001:**
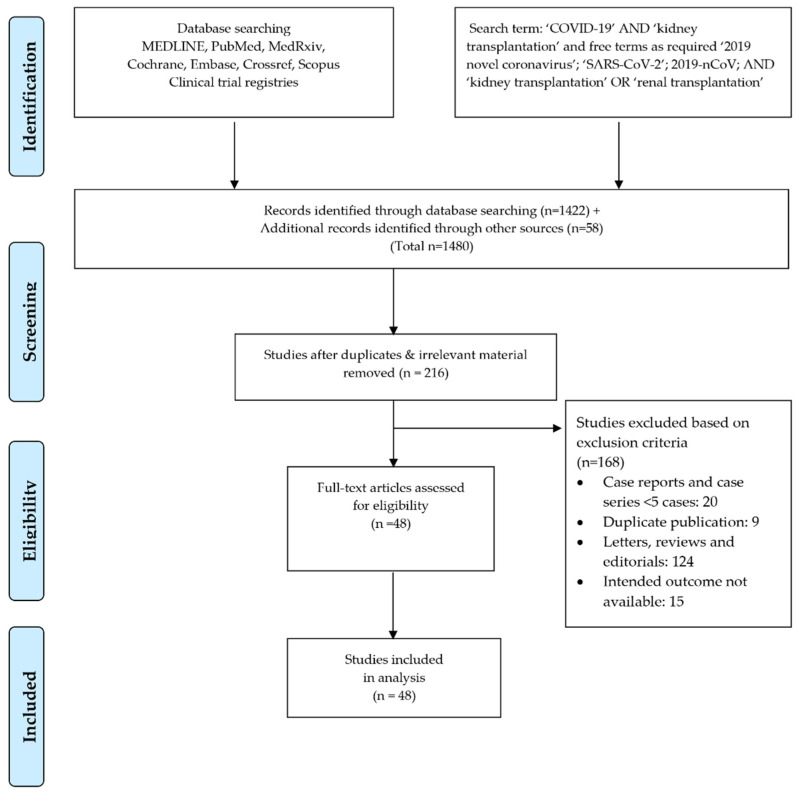
Search strategy and study selection used in this systematic review as per the PRISMA protocol.

**Figure 2 jcm-10-04533-f002:**
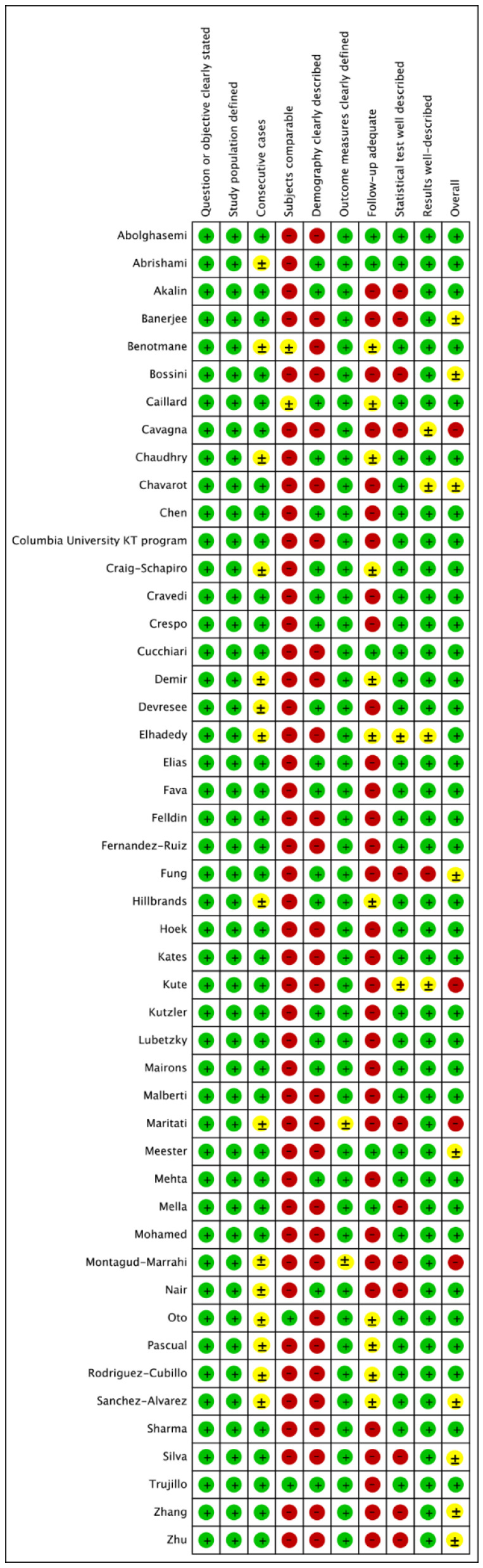
Quality assessment of included studies. (green-low risk of bias; yellow-unclear risk of bias; red-high risk of bias).

**Figure 3 jcm-10-04533-f003:**
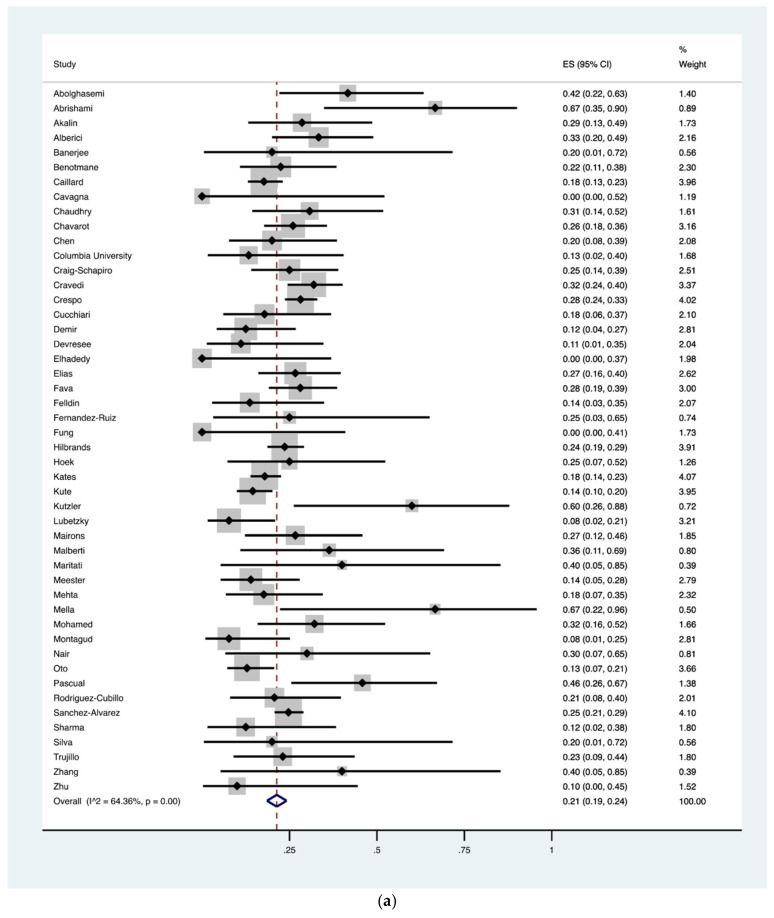
(**a**) Pooled prevalence of in-hospital mortality in kidney transplant recipients diagnosed with corona virus disease 2019 (COVID-19). In-hospital mortality was outlined in 718/3137 patients (48 studies). The vertical red dashed line represents the overall mean effect size of the studies (0.22) and prevalence of 22%. The edges of the blue diamond represent 95% confidence intervals (0.19–0.24); (**b**) Pooled prevalence of intensive care unit (ICU) mortality in kidney transplant recipients diagnosed with COVID-19. ICU mortality was outlined in 184/355 patients (24 studies). The vertical red dashed line represents the overall mean effect size of the studies (0.53) and prevalence of 53%. The edges of the blue diamond represent 95% confidence intervals (0.44–0.63); (**c**) Pooled prevalence of intensive care unit (ICU) admission in kidney transplant recipients diagnosed with COVID-19. ICU admission data were required in 570/2439 patients (37 studies). The vertical red dashed line represents the overall mean effect size of the studies (0.26) and a prevalence of 26%. The edges of the blue diamond represent 95% confidence intervals (0.22–0.30); (**d**) Pooled prevalence of acute respiratory distress syndrome (ARDS) in kidney transplant recipients diagnosed with COVID-19. ARDS was present in 197/344 patients as reported in 13 studies. The vertical red dashed line represents the overall mean effect size of the studies (0.58) and a prevalence of 58%. The edges of the blue diamond represent 95% confidence intervals (0.48–0.69); (**e**) Pooled prevalence of mechanical ventilation requirement in kidney transplant recipients diagnosed with COVID-19. Mechanical ventilation was needed in 433/1848 patients (33 studies). The vertical red dashed line represents the overall mean effect size of the studies (0.24) and prevalence of 25%. The edges of the blue diamond represent 95% confidence intervals (0.20–0.28); (**f**) Pooled prevalence of mortality while on invasive ventilation in kidney transplant recipients diagnosed with COVID-19. Mortality in patients requiring invasive ventilation was 206/285 as reported (24 studies). The vertical red dashed line represents the overall mean effect size of the studies (0.68) and prevalence of 68%. The edges of the blue diamond represent 95% confidence intervals (0.58–0.79).

**Figure 4 jcm-10-04533-f004:**
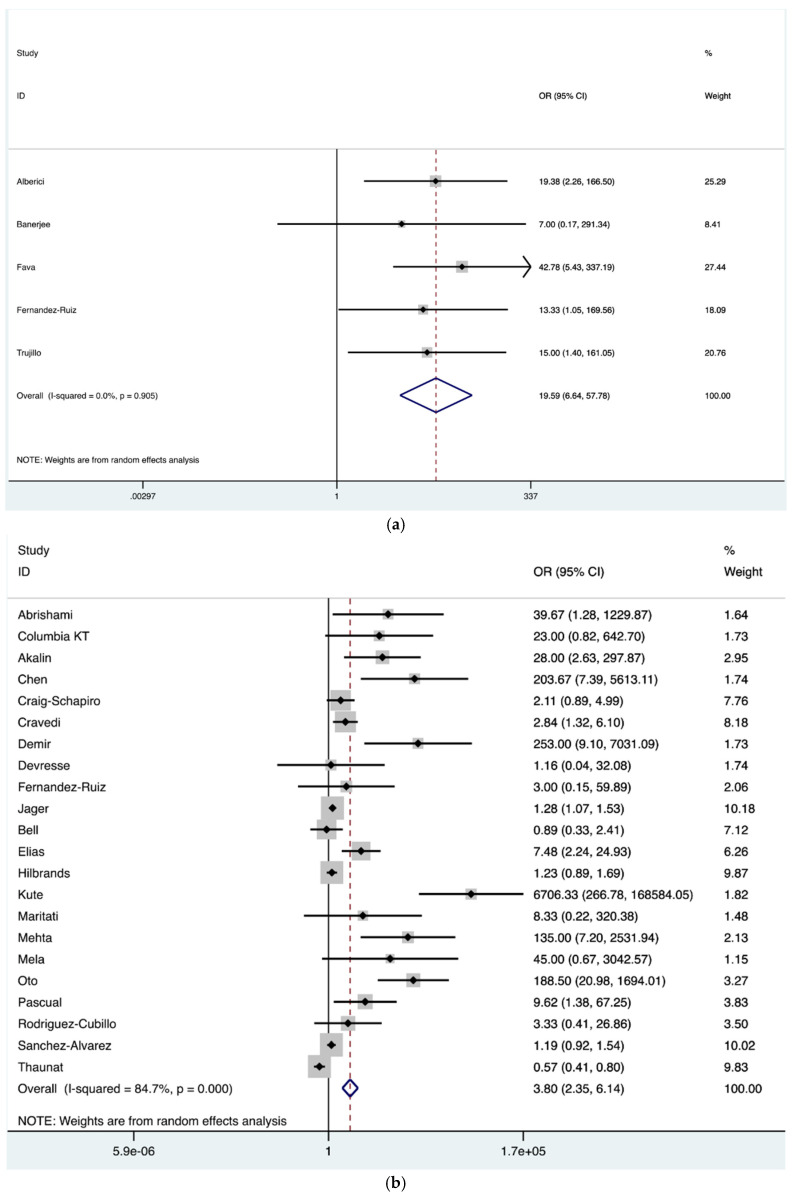
(**a**) Forest plot depicting COVID-19 associated mortality risk in kidney transplant recipients with ARDS in contrast to no ARDS. The diamond shows increased risk in patients with ARDS, OR = 19.59 (red dashed line). The edges of the blue diamond represent 95% confidence intervals (6.64–57.78). ARDS: Acute respiratory distress syndrome; (**b**) Forest plot depicting COVID-19 associated mortality risk in kidney transplant recipients on invasive ventilation in contrast to not on invasive ventilation. The diamond shows increased risk in patients receiving invasive ventilation, OR 3.80 represented by red dashed line with the edges of the blue diamond representing 95% confidence intervals (2.35–6.14); (**c**) Forest plot depicting COVID-19 associated mortality risk in kidney transplant recipients receiving critical care in contrast to hospital admission without critical care. The diamond shows increased risk in patients receiving critical care group, OR 13.39 represented by red dashed line with the edges of the blue diamond representing 95% confidence intervals (7.27–24.68). OR: Odds ratio.

**Figure 5 jcm-10-04533-f005:**
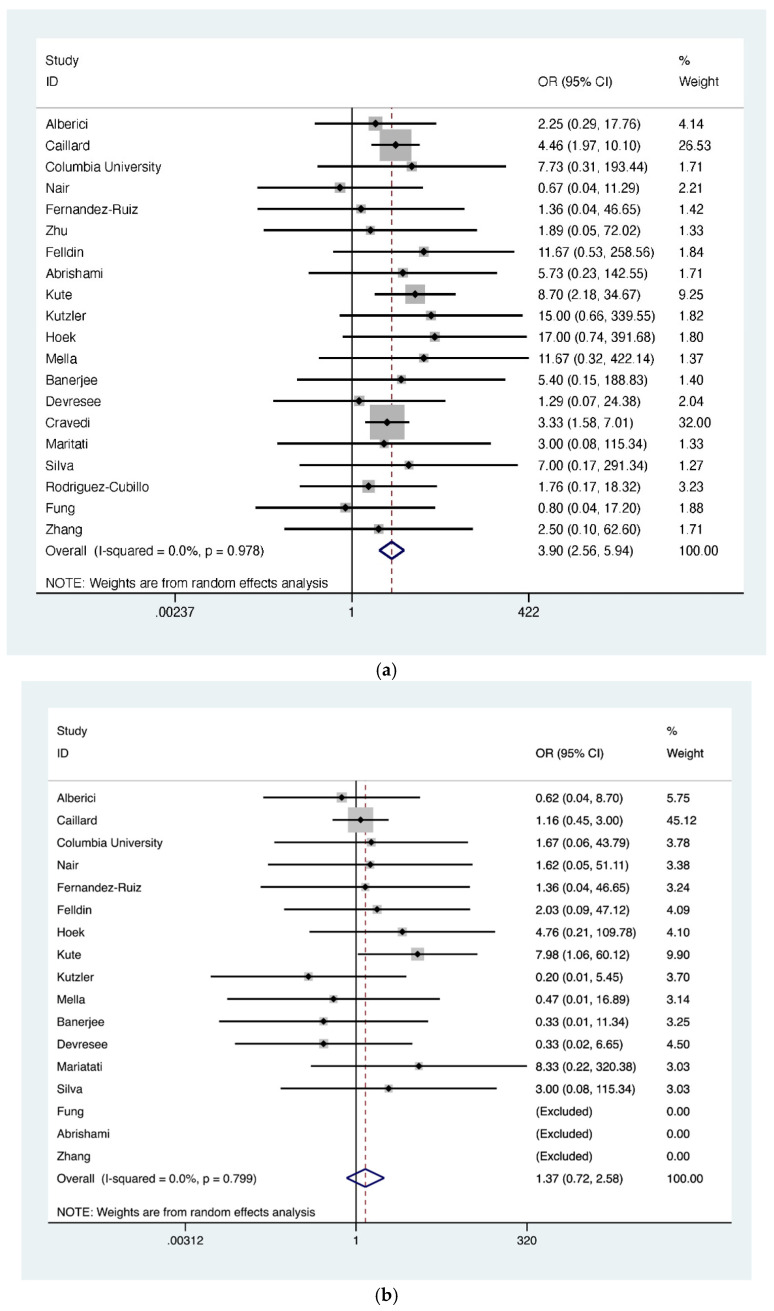
(**a**) Forest plot depicting COVID-19 associated mortality risk in kidney transplant recipients. The diamond shows higher risk in patient group ≥60 years of age, OR= 3.90 (red dashed line). The edges of the blue diamond represent 95% confidence intervals (2.56–5.94); (**b**) Forest plot depicting COVID-19 associated mortality risk in kidney transplant recipients in the late post-transplant period group (>2 years) in contrast to the early post-transplant period (≤2 years). The diamond shows no increased risk between the groups; OR 1.37 represented by red dashed line with the edges of the blue diamond representing 95% confidence intervals (0.72–2.58); (**c**) Forest plot depicting sex related mortality risk in COVID-19 suffering kidney transplant recipients. The diamond shows no increased risk between the groups; OR 0.71 represented by vertical red dashed line with the edges of the blue diamond representing 95% confidence intervals (0.50–1.01). OR: Odds ratio.

**Figure 6 jcm-10-04533-f006:**
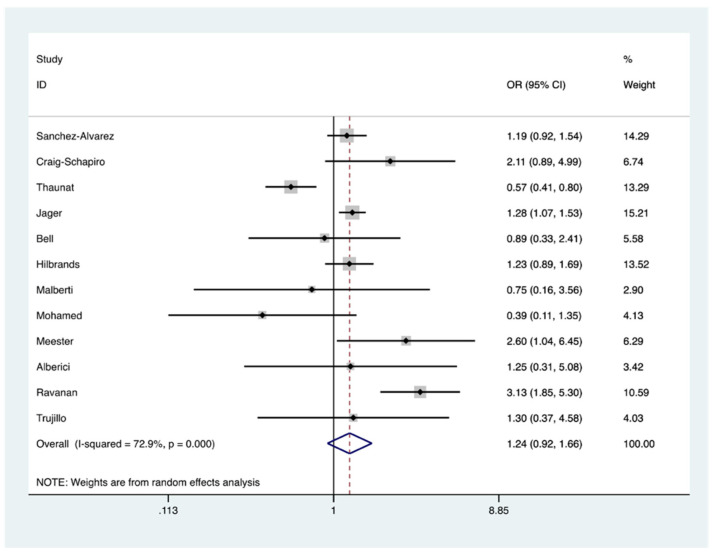
Forest plot depicting COVID-19 associated mortality risk in chronic kidney disease (CKD) patients on waitlist/dialysis in contrast to kidney transplant recipients. The diamond shows no increased risk of death; OR 1.24 represented by red dashed line with the edges of the blue diamond representing 95% confidence intervals (0.92–1.66). OR: Odds ratio.

## Data Availability

No new data were created in this study.

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
