# Peer review of "The Impact of COVID-19 on Kidney Transplant Recipients in Pre-Vaccination and Delta Strain Era: A Systematic Review and Meta-Analysis"

_jcm, 2021, doi:10.3390/jcm10194533_

Round 1
Reviewer 1 Report
The work by Jayant and colleagues is very well written and an in-depth analysis of the available evidence regarding COVID-19 in the kidney transplant population. I have no specific criticism for the Authors, which I would like to congratulate for their effort.
Author Response
Response 1. Thank you very much for this comment and time contributed for reviewing our Manuscript.
Reviewer 2 Report
This work is an extensive review of the early impact of the covid19 pandemic had on renal transplant recipients. The findings are very important because they show that renal transplant patients had similar mortality to other end stage renal disease patients. Papers for this study were published prior to April 2020 so this experience does not include effects from vaccination. Although this is included in the title it may be worth pointing out in the discussion. I found the age analysis confusing grouping patients above and below age 60-65. if a patient was 62 what group would they fall into? All in all this is a high quality work which serves as an important reference point for the ongoing covid pandemic.
Author Response
Response 2. Thank you very much for your invaluable comments.
- There were 23 studies, which provided outcomes stratified according to the age with cut off of 60 and only 3 studies with cut off of 65. We re-analyzed the outcomes again separately for each cut off and these are the results:
- a) Mortality risk in age ≥60 years
Death in ≥60 years 106 out of 322 compared to 59 out of 476 in <60 years; OR 3.90 (95% CI 2.56-5.94); P Value <0.000; Heterogeneity- 0%
- b) Mortality risk in age ≥65 years
Only 3 studies
Death in ≥65 years 87 out of 205 compared to 39 out of 276 in <65 years; OR 4.44 (95% CI 2.86-6.89); P Value <0.000; Heterogeneity- 0%
Therefore, we replaced previous combine outcome for cutoff 60 and 65 with outcomes only for cut off of 60 in the corrected manuscript for clarity with associated figure 5A (new Figure 5A in revised Manuscript and attached to this reply)
- In order to address additional comments, we added the following sentence in the discussion as suggested by the Reviewer (page 15 rows 436 - 441):
"….. At the same time, COVID-19 vaccines were not available and variant delta not described. Therefore, our study provides insight on the impact of Covid-19 on kidney transplant population in such conditions. Nevertheless, it provides valuable information and may serve as a reference for the future studies assessing impact of patient vaccination as well as delta variant of COVID-19 in kidney transplant population in the future."

Reviewer 3 Report
Nicely done, and great source of information for kidney transplant patient population and in congratulate you.
I suggest that if you add date of actual transplant, kidney function prior to diagnose with Covid and immunosuppression regimen, these make the paper more attractive!
thank you
Author Response
Response 3. Thank you very much for this comment. We agree with Reviewer’s suggestion. The requested information can be found in the Table 1A: (column #9) “Immunosuppression regimen”, mean time since the transplant was listed (column # 10) and baseline serum creatinine (median and range) in the last column (#11), unless such data was unavailable in the source article.